# Metabolomic Panel for the Diagnosis of Heart Failure with Preserved Ejection Fraction

**DOI:** 10.3390/ijms26052102

**Published:** 2025-02-27

**Authors:** Maria V. Kozhevnikova, Anastasiia V. Kakotkina, Ekaterina O. Korobkova, Ivan V. Kuznetsov, Ksenia M. Shestakova, Natalia E. Moskaleva, Svetlana A. Appolonova, Yuri N. Belenkov

**Affiliations:** 1Hospital Therapy No. 1 Department, N.V. Sklifosovskiy Institute of Clinical Medicine, I.M. Sechenov First Moscow Medical University (Sechenov University), 119991 Moscow, Russia; kozhevnikova_m_v@staff.sechenov.ru (M.V.K.); krivova_a_v@staff.sechenov.ru (A.V.K.); kuznetsov_i_v_2@staff.sechenov.ru (I.V.K.); belenkov_yu_n@staff.sechenov.ru (Y.N.B.); 2Centre of Biopharmaceutical Analysis and Metabolomics, Institute of Translational Medicine and Biotechnology, I.M. Sechenov First Moscow Medical University (Sechenov University), 117418 Moscow, Russia; shestakova_k_m@staff.sechenov.ru (K.M.S.); moskaleva_n_e@staff.sechenov.ru (N.E.M.); appolonova_s_a@staff.sechenov.ru (S.A.A.)

**Keywords:** chronic heart disease, hypertension, metabolites, metabolomics, metabolomic panel

## Abstract

The diagnosis of heart failure with preserved ejection fraction (HFpEF) remains challenging. The use of metabolomics approaches seems promising in speeding up and simplifying the diagnostic process in HFpEF patients, which can lead to earlier treatment initiation and better improvement of patient condition. The aim of this study was to develop a diagnostic panel of metabolites (metabolomic biomarkers) for the detection and diagnosis of HF with preserved ejection fraction. The study included 76 participants with hypertension, 36 of whom were diagnosed with HFpEF. The blood plasma metabolomic profile, including 72 metabolites, was detected using high-performance liquid chromatography combined with mass spectrometry. There were 18 statistically significant differences in concentrations of metabolites and 3 differences in their ratios between HFpEF and hypertension groups. The prognostic model for detecting the possibility of HFpEF included seven metabolites and two ratios: hexadecenoylcarnitine, arginine, trimethylamine-N-oxide, asymmetric dimethylarginine (ADMA), arginine/ADMA ratio, kynurenine, kynurenine/tryptophan, neopterin, and anthranilic acid. The area under the ROC curve was 0.981 ± 0.017. The resulting model was statistically significant (*p* < 0.001). The metabolomic panel could be considered as an addition to the present HFpEF laboratory diagnostic criteria for blood plasma analysis in clinical practice.

## 1. Introduction

Currently, timely diagnosis and treatment of heart failure (HF) remain relevant. The prevalence of HF is increasing worldwide [1,2]. The lifetime risk of HF has increased to 24%, and the prevalence is expected to rise by almost two times [3]. Despite advances in diagnostic techniques, verifying heart failure with preserved ejection fraction (HFpEF) remains challenging. The major reasons for the increase in HFpEF are the aging of the population and the increasing burden of comorbid conditions, such as hypertension, atrial fibrillation, diabetes, and obesity [4,5].

Today the diagnosis of HFpEF is based on a combination of several criteria, including the presence of clinical symptoms, structural and functional changes of the heart identified by echocardiography, and elevated levels of B-type natriuretic peptide (BNP) [6,7,8,9,10]. However, BNP level can be influenced by various factors, and while it is a highly sensitive indicator for detecting HF with reduced ejection fraction (HFrEF), its normal levels do not completely exclude the possibility of HFpEF. It is also important to note that the modern diagnostic protocol for HF is based on detection at the stage of pronounced clinical manifestations [9,10]. Treatment initiated at an earlier stage can significantly reduce the mortality of patients with HF, which remains extremely high and comparable to the mortality in cancer [6,7,8,9]. Difficulties in the diagnosis of HFpEF, as well as the lack of specific treatment methods, are most likely related to an incomplete understanding of the pathophysiological mechanisms of HFpEF. Myocardial hypertrophy, fibrosis, diastolic dysfunction, low-level systemic inflammation, and endothelial dysfunction, which are behind the development of congestive heart failure (CHF), develop as a consequence and are accompanied by impaired systemic and myocardial metabolism, which subsequently leads to a vicious cycle that causes the formation of CHF and contributes to its progression [11].

It is impossible not to mention the efforts to find new potential proteomic diagnostic markers for CHF. For example, soluble ST2 (sST2) and galectin-3 proteins were included in the 2013 ACCF/AHA recommendations on additional risk stratification for patients with acute and chronic heart failure [12]. So, in the 2013 study, researchers found that galectin-3 plays a significant role in myocardial fibrosis, which determines the transition from compensated to decompensated heart failure, thus highlighting the clinical importance of the correlation between galectin-3 and HFpEF development [13]. However, the studies are inconsistent, as some compare galectin levels with HFpEF occurrence, while others compare them to other diagnostic measures such as NT-proBNP levels and echocardiographic data. Additionally, galectin levels vary significantly among patients in different studies, making it challenging to draw conclusions about the prognostic significance of the biomarker.

Another potential biomarker for HFpEF is soluble ST2 (sST2). In one of the latest studies in 2020, it was found that sST2, along with NT-proBNP levels, could be used to predict HFpEF [14]. The study conducted in 2022 found that sST2 has a higher prognostic significance than NT-proBNP for achieving cardiovascular disease (CVD) endpoints [15]. The researchers attributed that sST2 variations were not linked with body mass index (BMI), renal function, or age. Despite this, there is the study that found sST2 levels as a marker either for an all-cause or non-CVD mortality among patients with HFpEF, while the NT-proBNP levels were significantly associated with CVD mortality in the same cohort [16]. However, galectin-3 and sST2 are not widely used in the world at the moment due to the wide range of their concentrations in the blood, which makes their use difficult. It brings us back to the question of searching for potential biomarkers of HFpEF.

Currently, there is an increasing body of research on the use of metabolomics to understand the pathophysiology, find potential biomarkers, and identify therapeutic targets. For example, in a large Framingham study, the metabolomic profile of 2336 study participants was determined, and the association between kynurenine, diacylglycerol, and leucine metabolites and remodeling parameters such as left ventricular end-diastolic size was demonstrated [17]. Another study demonstrated a significant association between circulating acylcarnitine levels and the level of myocardial hypertrophy with diastolic dysfunction parameters [18]. Thus, studying metabolomic profile changes is a promising direction for discovering new CHF and determining new points of application for therapy in this patient population.

We focused on the investigation of the metabolomic biomarkers that differentiate HFpEF patients from patients with risk factors without signs and symptoms of HF using high-performance liquid chromatography (HPLC) combined with mass spectrometry in order to create a diagnostic panel.

## 2. Results

The groups were comparable in gender (*p* = 0.569) and age (*p* = 0.085). HFpEF patients had a typical phenotype: hypertension (100%), diabetes (47.2%), atrial fibrillation (55.6%), chronic kidney disease (55.6%), and obesity (72.2%). There were no significant differences between the groups in most risk factors and comorbidities, including smoking (*p* = 0.089), dyslipidemia (*p* = 0.221), impaired glucose tolerance or diabetes mellitus (*p* = 0.568), history of coronavirus infection (COVID-19) (*p* = 0.747), brachiocephalic artery atherosclerosis (*p* = 0.569), and chronic kidney disease (*p* = 0.074). However, there were some differences in other factors, such as obesity (*p* = 0.009), hyperuricemia (*p* = 0.013), and history of acute cerebral circulatory failure (*p* = 0.023) (Table 1), which were more common in the HFpEF group.

There were significant differences in the types of complaints reported by the patients. In the group with HFpEF, there were significantly more complaints of dyspnea (*p* < 0.001) and edema of the lower extremities (*p* < 0.001) than in patients without HFpEF.

Patients with HFpEF and patients with hypertension had different treatments at the time of their inclusion in the study, as shown in Table 1. Patients were included in the study while on baseline therapy for at least 2 weeks.

The data from laboratory and instrumental examinations are presented in Table 2.

After performing targeted metabolomic analysis on 76 blood plasma samples, the data were analyzed using PCA; an unsupervised PCA model was assigned for a preliminary overview of the data as well as for outliers’ exclusion. Figure 1A represents the two-dimensional PCA separation of the two studied groups that did not get full separation of the analyzed objects. Further, in order to get a clearer presentation of the differences among two groups, the OPLS-DA analysis was performed. Figure 1B shows nearly clear separation of the analyzed groups with the predicted variance Q2 (Cum) and variance explained-R2X (Cum) equal to 0.58 and 0.92, respectively. To elucidate the most significant features (metabolites) that contributed to the model, the variable importance in the projection (VIP) scores were calculated for each metabolite. The metabolites with a VIP score equal to or higher than 1 were assigned as important. Additionally, we represented an S-plot, characterizing the relationship between loading vectors of the OPLS-DA model (Figure 1C). The loading vectors characterize the direction and strength of the relationship between the variables. The discriminant functions identify variables that mostly contribute to the separation between the desired groups. Variables that are located far from the origin of the plot are considered important. The list of these metabolites is represented in Table 3.

The study revealed significant differences in levels of 33 metabolites related to acylcarnitines, amino acids, metabolites of tryptophan catabolism, and metabolites of nitrogen metabolism. Three ratios were also found in the main group (Figure 2). Correlation analysis revealed relationships between metabolite concentrations and clinical, echocardiographic, and functional parameters (Figure 3). Multiple linear and variance analyses were performed to examine the effects of statistically significantly different risk factors and comorbidities between groups, such as BMI (*p* = 0.001) and obesity (*p* = 0.009), higher uric acid levels (*p* = 0.016) and hyperuricemia (*p* = 0.013), atrial fibrillation, and therapy. Body mass index had a significant effect on the levels of medium-chain acylcarnitines (*p* = 0.002), tyrosine (*p* = 0.019), histidine (*p* < 0.001), ADMA (*p* = 0.007), antranillic acid (*p* = 0.010), and arginine (*p* < 0.001). C5-1 (*p* = 0.002) and Arginine/ADMA ratio (*p* = 0.006) were associated with uric acid metabolism disorders. Drug therapy (diuretics) had little effect on the kynurenine/tryptophan ratio (*p* = 0.047).

Higher levels of kynurenine (r = 0.445, *p* = 0.007) were associated with higher BMI; higher levels of hexadecenoylcarnitine (C16:1) (r = 0.505, *p* = 0.002), ADMA (r = 0.377, *p* = 0.023), anthranilic acid (r = 0.422, *p* = 0.010), neopterin (r = 0.379, *p* = 0.023) were correlated with higher NT-proBNP; lower levels of arginine (r = −0.526, *p* = 0.012) and TMAO (r = −0.563, *p* = 0.006) were associated with inflammatory markers with acceleration of ESR; higher levels of anthranilic acid (r = 0.426, *p* = 0.038), kynurenine (r = 0.566, *p* = 0.004) and kynurenine/tryptophan ratio (r = 0.459, *p* = 0.024) were associated with elevated C-reactive protein (CRP) levels; levels of tryptophan catabolism metabolites (anthranilic acid r = 0.454, *p* = 0.007, kynurenine r = 0.713, *p* < 0.001, kynurenine/tryptophan ratio r = 0.564, *p* < 0.001) and neopterin (r = 0.380, *p* = 0.026) were associated with uric acid levels; neopterin levels (r = 0.371, *p* = 0.026) were associated with creatinine levels; lower levels of C16:1 (r = −0.343, *p* = 0.040), arginine (r = −0.354, *p* = 0.034) and higher levels of kynurenine (r = 0.384, *p* = 0.021) correlated with glucose levels; lower levels of C16:1 correlated with lipid profile indicators, such as total cholesterol (r = −0.373, *p* = 0.030), LDL (r = −0.395, *p* = 0.028), VLDL (r = −0.536, *p* = 0.002), triglycerides (r = −0.438, *p* = 0.012); higher levels of TMAO were associated with HDL levels (r = 0.435, *p* = 0.016); lower levels of arginine/ADMA ratio were correlated with LV PW size, LVMW, LV and RA volume (r = −0.414, *p* = 0.012; r = −0.392, *p* = 0.018; r = −0.369, *p* = 0.029; r = −0.402, *p* = 0.015, respectively).

Receiver operating characteristic (ROC) curve analysis showed that 7 metabolites and 2 ratios (Table 4) were potentially sensitive and specific for the diagnosis of HFpEF (Table 5).

However, the evaluation of a single specific metabolite may not provide fully informative results because the concentrations of individual metabolites can vary in different diseases and conditions. Therefore, the selected metabolites were combined into a panel for further analyses.

We developed a prognostic model using binary logistic regression to assess the diagnostic potential of this panel of metabolites. This model predicted the occurrence (or outcome) of HFpEF based on a combination of these metabolites. The model equation describes the relationship between the probabilities of the presence or absence of HF depending on the specific combination of metabolites in the panel. The equation for this model is as follows:P = 1/(1 + e^−z^) × 100%(1)
$$\begin{array}{ll} z=-32.636 & -31.380\times C1-0.232\times C2+24.914\times C3\\ & +0.475\times C4+0.049\times R1+0.349\times C5\\ & +0.025\times C6+0.146\times C7+618.859\times R2 \end{array}$$
where P is the probability of having HFpEF, e is the base of the natural logarithm, *C*1 is the concentration of acylcarnitine C16:1 in plasma, *C*2 is the concentration of the amino acid arginine in plasma, *C*3 is the concentration of ADMA in plasma, *C*4 is the concentration of TMAO in plasma, *R*1 is the ratio of arginine/ADMA concentrations, *C*5 is the concentration of anthranilic acid in plasma, *C*6 is the concentration of kynurenine in plasma, *C*7 is the concentration of neopterin in plasma, and *R*2 is the ratio of kynurenine/tryptophan concentrations. The z-indices were calculated through the development of the logistic regression classification model using the Python SkLearn 16.0 package. For this purpose, the machine learning model was built using a training dataset (70% of the data). The test dataset (30%) was utilized for testing the model and assessment of its quality metrics.

The regression model obtained was statistically significant (*p* < 0.001). Based on the Nigelkerk coefficient of determination, the model explains 87.4% of the variance in the “HFpEF” indicator. To assess the relationship between the probability of HFpEF and the logistic function P, we performed ROC analysis and obtained the following curve (Figure 4): The area under the ROC curve was 0.981 ± 0.017, with a 95% confidence interval (CI) of 0.948–1.000. The resulting model was statistically significant (*p* < 0.001). The threshold value of the logistic function P in the cutoff point corresponding to the highest Youden index was 64.7%. The presence of HFpEF was predicted when the *p*-value was greater than or equal to 64.7%. The sensitivity and specificity of our model were 94.4% and 100%, respectively. Compared to BNP as a single predictor, our model has better specificity (1.00 vs. 0.83) and sensitivity (0.94 vs. 0.80).

The obtained regression model is statistically significant (*p* < 0.001).

Notably, the combination of several metabolites in the panel allows for greater stability in the diagnosis of HFpEF, as opposed to the use of a single metabolite as a biomarker, which can be highly variable.

## 3. Discussion

In our study, we compared patients with similar risk factors for HFpEF, such as age, obesity, dyslipidemia, hypertension, atrial fibrillation (AF), type 2 diabetes mellitus (DM), and chronic kidney disease. Since the cardiac myocardium is an energy-consuming organ, it is sensitive to metabolic changes. Metabolic maladaptation and metabolic disorders such as those caused by type 2 DM, obesity, and AF play an important role in CVD development. Thus, we decided to study changes in the metabolomic profile of HFpEF patients.

In our present research, given the lack of significant differences in sex and age between groups of patients with arterial hypertension (AH) and CHF, the possible difference in metabolites between these two groups can be explained by the fact that patients were at different stages of cardiovascular disease at the time they were included in the study. As a result, when comparing metabolomic profiles between patients with CHF and patients with AH, differences were found in levels of plasma concentrations of amino acids such as threonine, arginine, and histidine; nitrogen metabolites such as SDMA, ADMA, and TMAO; arginine/ADMA ratios; metabolites derived from the catabolism of tryptophan such as kynurenine and 3-hydroxykynurenine; pterins such as biopterin and neopterin; as well as long-chain acyl carnitines such as C16:1, C16-OH, and C18-OH.

Confirmation of impaired fatty acid oxidation was also demonstrated in our work in the form of increased concentration of circulating long-chain acylcarnitine C16:1 in patients with HFpEF compared with patients with AH, indicating a persistent shift of metabolomic remodeling toward glycolysis and deterioration of myocardial compensatory mechanisms. These changes are consistent with a number of other experimental studies of HF models and clinical studies of changes in fatty acid and acylcarnitine metabolism in obesity, AH, and coronary heart disease, in which the association of circulating acylcarnitines, and especially long-chain acylcarnitines, with unfavorable clinical outcomes was demonstrated [18,19,20].

In our work, we demonstrated a significant decrease in the level of arginine and an increase in the ADMA level in patients with HFpEF compared to patients with AH. The arginine/ADMA ratio was also significantly reduced when comparing the two groups. These results most likely indicate a progressive impairment of endothelial function in patients with heart failure. There was a correlation between the concentration of arginine, the arginine/ADMA ratio, and structural and functional changes in the myocardium of patients with CHF. In turn, a high level of ADMA correlated positively with an increased level of NT-proBNP and a decreased level of arginine with ESR, which confirms the significant contribution of disturbed arginine metabolism to the development and progression of CHF. Our data are consistent with experimental studies on an experimental model of congestive heart failure, where an increase in ADMA levels and a decrease in endothelial-dependent relaxation stimulated by acetylcholine were demonstrated. These findings are similar to those of several clinical studies that also demonstrate impaired NO metabolism in HFpEF [21,22].

In our study, significantly higher plasma TMAO concentrations were observed in patients with HFpEF compared to patients with AH. Recently, there has been an increasing number of papers on the association between TMAO and CVDs, such as atherosclerosis, AH, and MI [22]. Particularly interesting is the direct effect of TMAO on the development of myocardial hypertrophy and fibrosis. This mechanism was well described in an experimental study by Li et al., in which rats with induced transverse aortic constriction and pressure-overload-induced HF showed elevated levels of circulating TMAOs, which promoted myocardial hypertrophy via the Smad3 signaling pathway [23]. Thus, in one study, elevated levels of TMAO were shown to have prognostic significance in predicting cardiovascular events in patients with HFrEF, but not in those with HFpEF. In another study, TMAO levels contributed to the risk stratification of HFpEF patients, especially when BNP levels were low [24,25]. However, this does not rule out the possibility that all described pathophysiological mechanisms of TMAOs influence represent a vicious circle that worsens the course of HF. In our study, higher levels of TMAOS were correlated with lower levels of ESR and were not associated with CRP. The revealed direct correlations between the metabolite and myocardial remodeling parameters (LVMI) support the idea that impaired TMAO metabolism contributes to CHF development.

Another interesting result of our work is the detection of increased plasma neopterin levels in patients with HFpEF. They are derivatives of pterins, the increased concentrations of which are characterized by tetrahydrobiopterin (BH4) deficiency. BH4 is a cofactor of a number of important biochemical reactions occurring in the body. BH4 deficiency leads to inhibition of such reactions as tyrosine formation from phenylalanine, dopamine and serotonin synthesis, NO formation from arginine, and oxidative cleavage of lipid esters [26]. Consequently, the increased level of pterin derivatives in our study impairs NO bioavailability and partially explains the decreased tyrosine level in patients with CHF compared to patients with AH.

The changes occurring in the kynurenine pathway of tryptophan catabolism in our study cannot be overlooked. A significant increase in the products of this pathway (kynurenine, 3-hydroxykynurenine, anthranilic acid, and 3-hydroxanthranilic acid) and a change in the ratio of kynurenine/tryptophan were observed. Among other things, correlations were found between CRP levels and kynurenine, and between CRP and the kynurenine/tryptophan ratio. Kynurenines are known to be associated with low-level systemic inflammation in the body, and their predominance in the catabolic pathway contributes to decreased concentrations of niacin. This can positively affect carbohydrate metabolism, lipid profile, and microcirculation (vasodilatation) and promote cellular energy homeostasis through the formation of NAD+, a common redox cofactor involved in various biological processes [27,28]. The present study demonstrated progressive impairment of tryptophan catabolism in patients with CHF, which may suggest greater severity of low-level systemic inflammation contributing to further progression of the disease.

In accordance with the obtained data, we proposed and analyzed a diagnostic panel of metabolites. The proposed method for HFpEF diagnosis involves determining the most significant metabolites in blood plasma, including arginine, ADMA, TMAO, the arginine/ADMA ratio, kynurenine, anthranilic acid, the kynurenine/tryptophan ratio, neopterine, and acylcarnitine C16:1 as biomarkers of HFpEF.

Our results are comparable with other experimental and clinical studies, the authors of which have also proposed metabolite detection and metabolomic panels for the diagnosis of CHF, especially HFpEF [18,29,30,31,32]. In an experimental study, all CHF phenotypes were analyzed by EF, and a mouse model of HFpEF was obtained, for which one hundred and nine differentially expressed metabolites were detected. When compared with controls, the most significantly altered metabolites were glycerophospholipids [29]. Thus, a group of researchers led by Christian Delles analyzed the metabolic profiles of more than 10,000 patients participating in the PROSPER and FINRISK trials. Among them, 315 were hospitalized with decompensated HF. They found an association between phenylalanine levels and the risk of HF [30]. They also identified phenylalanine as a potential predictor of hospitalization for CHF. However, a limitation of this study was the determination of the metabolic profile using nuclear magnetic resonance (NMR) spectroscopy on blood samples collected approximately 20 years ago. Such a long storage period might lead to the degradation of some metabolic compounds [30]. Another large-scale study was conducted by Charlotte Andersson et al., who analyzed 217 circulating metabolites in the blood of 2336 participants of the Framingham study who initially had no CHF [18]. During the follow-up period, 219 participants were diagnosed with CHF, with a mean time to diagnosis of 12.6 years. Lower levels of phosphatidylcholine and lysophosphatidylcholine and higher levels of ADMA were associated with a higher risk of CHF development [18]. However, blood samples were collected 20 years before diagnoses, and the study had limitations, including the fact that the diagnostic criteria for CHF at the time of follow-up (1976–2016) did not include echocardiography data. One of the few similar studies in this area is the study by Marcinkiewicz-Siemion et al., which attempted to test the feasibility of a combined approach using nontargeted metabolomics (liquid chromatography-mass spectrometry) and machine learning to create a diagnostic panel for HFrEF [31]. Patients with CHF (*n* = 67) were included in the study, whereas the comparison group consisted of patients without signs of CHF (*n* = 39). During statistical processing of the results, eight metabolites were selected for the study: uric acid, two isomers of long-chain acylcarnitines (C18:2 and C20:1), deoxycholic acid, docosahexaenoic acid, and one unknown metabolite. These metabolites have demonstrated prognostic value in patients with CHF, and the accuracy of the panel was comparable to that of BNP levels [31]. There was another study where Wang et al. evaluated the prognostic significance of amino acids in 712 patients with CHF at different stages [32]. Based on those results, the authors proposed a panel containing histidine, ornithine, and phenylalanine as an alternative for NT-proBNP [32]. Unfortunately, both of these studies included the determination of metabolites only in patients with LVEF less than 50%. In the modern world, the increasing contribution of HFpEF to the structure of CHF is recognized, and the interest in this phenotype is increasing because of the difficulties in its detection and diagnosis.

Despite the large number of studies conducted, metabolomic panels and the determination of various metabolites have not been included in clinical guidelines for CVD. This may be due to the lack of unified approaches, analytical methods, or separation techniques that could provide a wide range of results [33,34,35]. Through targeted metabolomics, our studies have demonstrated that the plasma metabolomic profile of patients with HFpEF can be used to create a diagnostic panel that may serve as an additional or alternative method for verifying HFpEF, compared to natriuretic peptides. However, more extensive studies are needed to fully investigate this potential.

## 4. Materials and Methods

To identify potential HFpEF markers, a one-stage observational study was conducted at the I.M. Sechenov First Moscow State Medical University of the Russian Ministry of Health (Sechenov University).

This study included 76 participants with CVD: 36 HFpEF patients and 40 patients with hypertension. HF was diagnosed based on the criteria set by the European Society of Cardiology and the Russian Society of Heart Failure Specialists [9,10]. Hypertension was defined based on repeated office systolic blood pressure (SBP) values of 140 mmHg and/or diastolic blood pressure (DBP) of 90 mmHg [36]. Clinical examination, measurement of the level of the N-terminal propeptide of B-type natriuretic peptide (NT-proBNP), and echocardiographic examination were used to exclude the presence of HF in the comparison group. Diagnosing obesity was established as a body mass index (BMI) ≥ 30 kg/m^2^. The diagnosis of dyslipidemia was established based on an increase in LDL, triglycerides, total cholesterol, or a decrease in HDL, according to current recommendations or the use of lipid-lowering therapy. The diagnosis of hyperuricemia was established based on an increase in the level of uric acid >416.5 μmol/L for men and >339.2 μmol/L for women. Damage to the brachiocephalic arteries was established based on the detection of the presence of atherosclerosis of the brachiocephalic arteries according to the ultrasound Doppler. The diagnosis of impaired glucose tolerance and diabetes was based on the level of glycated hemoglobin ≥ 6.5% and venous blood glucose > 7.0 mmol/L and 7.8 < 11.1 2 h after an oral glucose tolerance test or taking hypoglycemic drugs. Chronic kidney disease was established in the presence of any clinical signs indicating kidney damage and persisting for at least three months AND/OR a decrease in SCF < 60 mL/min/1.73 m^2^ persisting for three or more months, regardless of the presence of other signs of kidney damage.

The exclusion criteria were the same for both groups: presence of HFrEF, HFmrEF, ischemic heart disease, secondary arterial hypertension, cardiomyopathies, congenital cardiac malformations, acute inflammatory conditions, strokes, transient ischemic attacks within the previous six months, severe liver or kidney dysfunction, bronchial asthma, chronic obstructive pulmonary disease, exacerbation of gastrointestinal disorders, malignant neoplasms, endocrine disorders in the medical history, autoimmune diseases, psychiatric disorders, alcohol abuse or narcotic substance use, pregnancy, and lactation.

### 4.1. Clinical Parameters

On the first day of participation in the study, in all patients we evaluated their complaints, medical history, clinical and demographic information: weight, height, BMI, SBP and DBP, heart rate (HR); New York Heart Association functional class (NYHA); fasting laboratory tests: total blood count, lipid profile, uric acid level, iron level, renal function tests, NT-proBNP level, CRP and echocardiographic parameters: left ventricular ejection fraction (LVEF), interventricular septum (IVS), posterior wall (PW) thickness, end-diastolic dimension (ED), left atrial volume (LAV), right atrial volume (RAV), pulmonary artery systolic pressure (PASP), the ratio of maximum velocities of early filling transmittal blood flow and movement of the fibrous ring in early diastole (E/e’), the ratio of transmittal flow indices in the phases of early and late diastole LV filling (E/A), as well as indices such as left atrial volume index (LAVI), relative wall thickness (RWT), left ventricular myocardial mass (LVMW), and left ventricular myocardial index (LVMI), were calculated. Functional assessment was also performed using the six-minute walk test (6 MWT).

Concentrations of the N-terminal fragment of the brain natriuretic peptide precursor (NT-proBNP) were measured in serum using a reagent kit for immunoenzymatic determination of the concentration of NT-proBNPs (Cloud-Clone Corp., Katy, TX, USA). Echocardiographic parameters were assessed using two-dimensional echocardiography in M-mode and B-mode, pulsed Doppler imaging, and continuous Doppler in the supine position with the Vivid 7 Dimension/Vivid 7 PRO echocardiographs version 6.0 (General Electric Co., Horten, Norway).

### 4.2. Identification of Metabolites

Blood samples were collected from all patients on the first day of participation in the study, in the morning, after an overnight fast, in tubes containing dehydrated disodium salt of ethylenediaminetetraacetic acid. The samples were centrifuged at 2000 RPM for 20 min, and the resulting plasma was stored at −80 °C until analysis. After collecting all biological materials, targeted metabolomic profiling of the blood plasma from all participants was performed using a Waters Acquity UPLC system coupled with a high-resolution mass spectrometer (TSQ, Xevo TQ-S Micro, Waters, Milford, MA, USA). We analyzed 76 plasma samples and determined the metabolic profile of each patient, including 72 different metabolites such as amino acids, metabolites from tryptophan catabolic pathways, nitrogen metabolism products, acylcarnitines, and pterin derivatives. The metabolites for targeted metabolomic analysis were selected based on those previously identified in the preliminary untargeted metabolomic experiments as well as the comprehensive literature review.

None of the patients included in the study made any changes to their diet or took any vitamins or dietary supplements three days before blood collection. None of the patients followed a vegetarian diet during the study.

Quantitative analysis of amino acids (AA) and acylcarnitines (AC) was performed by liquid chromatography with mass-spectrometric detection. For the sample preparation, a 10 μL aliquot of each plasma sample was mixed with 50 μL of ISTD mix solution from the MassChrom Amino Acids and Acylcarnitines Non Derivatized 57000 Kit (Chromsystems, Gräfelfing/München, Germany) and 40 μL of methanol for protein precipitation and was centrifuged for 5 min at 100× *g.* 40 μL of the received supernatant was transferred into an LC-MS vial, diluted with 40 μL of water, and was ready for the consequent LC-MS/MS analysis.

The instrumental analysis was made using the Waters Acquity I HPLC system coupled to the Waters TQ-S-micro triple quadrupole mass spectrometer (Waters Corp, Milford, MA, USA). Chromatographic separation was achieved using a Waters ACQUITY BEH C18 column 1.7 μm, 100 mm × 2.1 mm (Waters, Milford, MA, USA). Mobile phases consisted of LC-MS grade water with 0.1% formic acid (phase A) and acetonitrile containing 0.1% formic acid (phase B). Flow rate was set at 0.3 mL/min. The column temperature was maintained at 40 °C. The gradient program was as follows: 1 min—1% B, 3 min—20% B, 5 min—90% B, 8 min—90% B, 8.1 min—1% B, 12 min—1% B. A mass spectrometric detector with a triple quadrupole in MRM mode was used for analysis (Appendix A). Mass spectrometric conditions were as follows: dwell time 0.019–0.025 s; capillary voltage—2 KV; collision gas medium—nitrogen; source temperature—150 °C. Preprocessing and data import were performed in TargetLynx software v4.2 (Waters, Milford, MA, USA).

Sample preparation for profiling the tryptophan catabolites was performed as follows: 100 µL of plasma (calibrators or QCs) was mixed with the internal standard (10 µL of stock solution of 10 µg/mL solution of 2-hydroxynicotinic acid) and 400 µL of acetonitrile. The mixture was then vortexed, centrifuged for 10 min at 13,000 rpm, and evaporated to dryness in a vacuum centrifuge evaporator at 37 °C. The residues were further reconstituted with 100 µL of a solution of 0.02% ascorbic acid in 10% methanol, centrifuged, and transferred into the LC-MS vial. Five µL of the extract was injected into the liquid chromatograph for the subsequent LC-MS/MS analysis.

Instrumental analysis was made using an Agilent 1200 liquid chromatograph with a 6450C tandem mass spectrometer (Agilent Technologies, Palo Alto, CA, USA). The chromatographic separation was achieved on a Discovery PFP HS F5 2.1 × 150, 3 µm column (Supelco Inc., Bellefonte, PA, USA) equipped with a Waters WAT084560 guard column (Waters Inc., Milford, MA, USA). The column temperature and the flow rate were set at 40 °C and 0.4 mL/min, respectively. Mobile phases consisted of 0.1% formic acid aqueous solution (phase A) and acetonitrile (phase B). The gradient program was as follows: 0 min—1% B; 4 min—10% B; 9 min—90% B; 10 min—90% B; 10.1—1% B; 12 min—1% B. Electrospray ionization was operated in positive mode. Main MS parameters were as follows: gas temperature—300 °C; gas flow—8 L/min; nebulizer gas—20 psi; sheath gas heater—300; sheath gas flow—10 L/min; capillary voltage—3500 V. Analytes were detected using MRM transitions presented in Appendix A.

Sample preparation for ADMA and SDMA determination was performed as follows: A 10 μL aliquot of each plasma sample (calibrator or QC sample) was mixed with 50 μL of isotopically labeled internal standards (ISTD) solution (D7-Arg, 1.55 μM) and 40 μL of methanol for protein precipitation. After a 10 min incubation, the sample was centrifuged for 5 min at 100× *g*. Further, 40 μL of supernatant was transferred to a vial, mixed with 40 μL of water, and 1 μL of the received solution was injected into the LC-MS/MS system.

Samples were analyzed using the Waters Acquity I HPLC system, coupled to the Waters TQ-S-micro triple quadrupole mass spectrometer (Waters Corp, Milford, MA, USA). Chromatographic separation was conducted using a Waters ACQUITY BEH C18 column 1.7 μm, 100 mm × 2.1 mm (Waters, USA). Mobile phases consisted of water with 2 mM ammonium formate and 0.015% heptafluorobutyric acid (mobile phase A) and methanol, containing 2 mM ammonium formate and 0.015% heptafluorobutyric acid (phase B). The gradient program was as follows: 1% B at 1 min, 20% B at 5 min, 90% B at 6 min, 90% B at 11 min, 1% B at 11.1 min, and 1% B at 15 min. The flow rate and column temperature were 0.3 mL/min and 40 °C, respectively. Mass-spectrometric (MS) detection was achieved by multiple reactions monitoring (MRM) (Appendix A) with a dwell time of 20 ms. Capillary and cone voltages were 1 and 19 V, respectively. Source temperature was set at 150 °C, desolvation temperature at −10 L/min; source and desolvation gas flow rates were 10 L/min and 1000 L/h, respectively.

Methods were validated for selectivity, linearity, precision, accuracy, recovery, matrix effect, and stability according to US FDA and EMA guidelines for bioanalytical method validation (EMA, 2019; USFDA, 2018). Detection limit and linear range are selected for each compound according to physiological plasma concentrations.

### 4.3. Statistical Analysis

Principal component analysis (PCA) was performed using the Python programming language. Multivariate analysis was carried out by the orthogonal partial least squares-discriminant analysis (OPLS-DA) model using SIMCA 13.5 software. Based on the results, metabolites with the highest VIP scores were identified. In addition, metabolites with a significant impact on the model were further assessed using the S-plot function.

The heatmap was created using Metaboanalyst 6.0 (xia-lab, University of Alberta, Edmonton, AB, Canada) and represents a graphical depiction of the average concentrations of the investigated metabolites from the main and comparison groups.

Statistical processing of the data collected in the program was performed using STATTECHv4.0 software. Arithmetic mean (M) and standard deviation (SD) were used to describe the normal distribution of quantitative indicators. Median (Me), lower and upper quartiles [Q1;Q3] were used for indicators with non-normal distributions. Qualitative indicators are presented as absolute numbers (*n*) and percentages (%). The distribution type of quantitative indicators was assessed using the Shapiro-Wilk test, as the number of participants in each group was less than 50.

In case of a normal distribution of quantitative indicators in two groups being compared, if the dispersions were equal, the Student’s t-test was used. However, if the dispersions were not equal, the Welch’s *t*-test was employed. If the distributions of the quantitative indicators in the compared groups were not normal, the Mann–Whitney U test was utilized. If the *p*-value was ≤ 0.05, the difference was deemed to be statistically significant; ≤0.01, very significant; and ≤0.001, maximally significant.

Pearson’s correlation coefficient was used in case of a normal distribution of indicators, and Spearman’s in case of other distributions. The Cheddock scale was used to assess the strength of correlations: 0–0.3 was very weak, 0.3–0.5 was weak, 0.5–0.7 was medium, 0.7–0.9 was high, and 0.9–1.0 is very high. When there was a negative correlation, the value of the relationship was reversed.

A predictive model for the probability of a specific outcome was constructed using logistic regression. The Nagelkerke R-squared coefficient serves as an indicator of certainty, reflecting the proportion of variance explained by the logistic regression model. ROC curve analysis was used to evaluate the diagnostic value of quantitative traits in predicting a particular outcome. The optimal cut-off point for a quantitative feature was determined by selecting the highest Youden index value. The metabolomic profile dataset was processed using Metaboanalyst 6.0 (xia-lab, University of Alberta, Edmonton, AB, Canada).

### 4.4. Limitations

A limitation of our study was the small number of participants in each subgroup. However, this was to ensure selecting the most homologous group of HFpEF using strict inclusion and exclusion criteria. Additionally, the differences in drug therapy between the main and control groups cannot be ignored. Larger studies with prospective follow-up groups are necessary for clinical validation of the diagnostic panel.

## 5. Conclusions

Metabolomic analysis of patients with hypertension and with HFpEF, developed as its complication, was performed. HFpEF presence was associated with increased levels of medium- and long-chain acylcarnitines, ADMA, TMAO, and metabolites of the kynurenine pathway of tryptophan metabolism. Metabolite levels were also matched with clinical and laboratory parameters, characterizing the severity of the disease. Data from metabolomic profiling, combined with ROC analysis, allowed us to develop a panel of metabolites to assess the likelihood of the presence of HFpEF. These study outcomes may lead to the development of novel methods of diagnosing HFpEF in the daily clinical practice of physicians of all specialties.

## Figures and Tables

**Figure 1 ijms-26-02102-f001:**
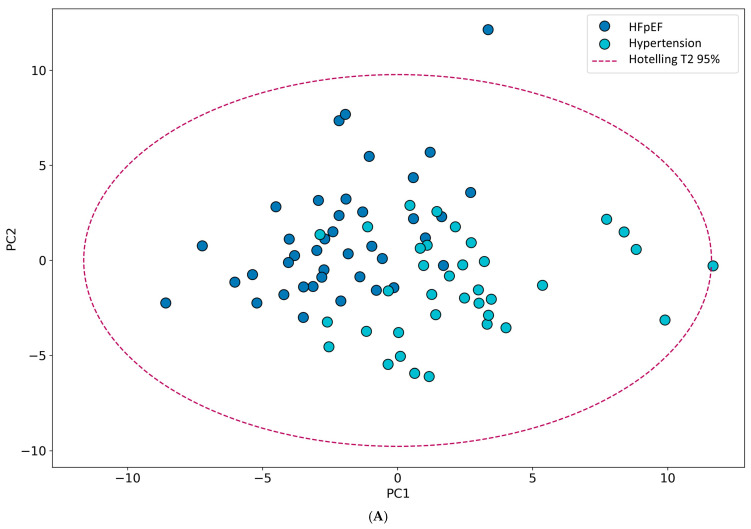
Differences in the concentrations of metabolites between HFpEF and hypertension groups. (**A**) two-dimensional PCA separation of the two studied groups; (**B**) differences between two groups using the OPLS-DA method; (**C**) S-plot of the model.

**Figure 2 ijms-26-02102-f002:**
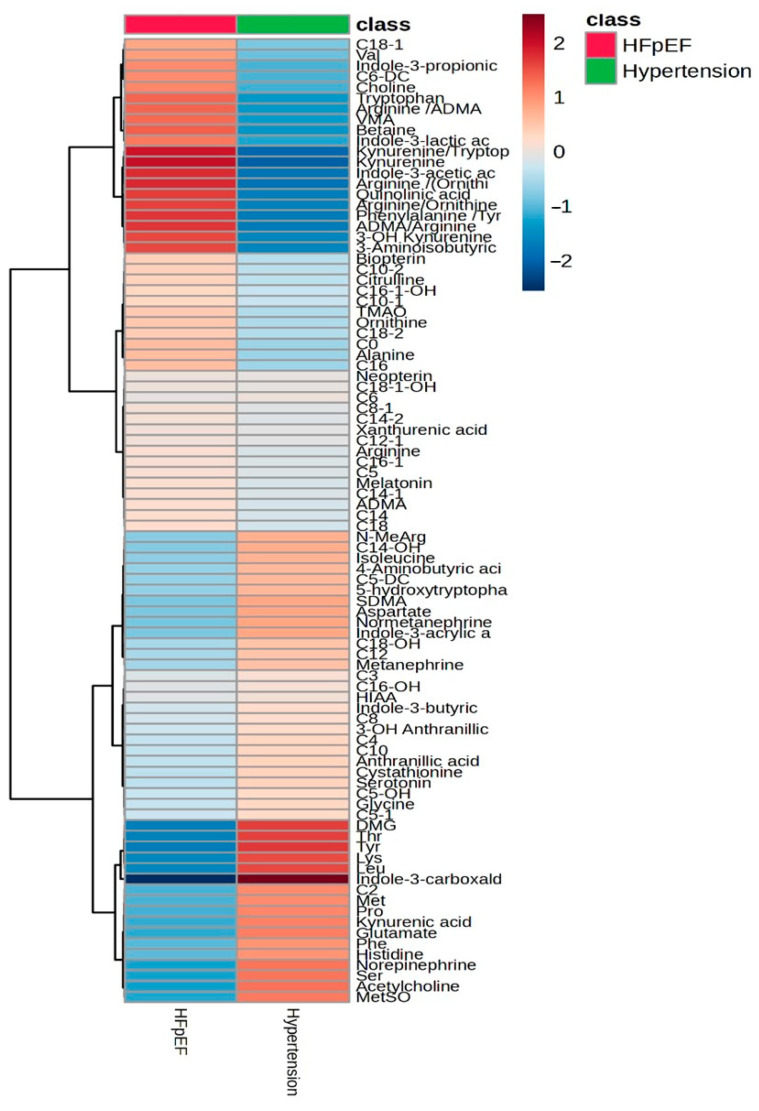
Differences in the concentrations of metabolites between HFpEF and hypertension groups using a heatmap.

**Figure 3 ijms-26-02102-f003:**
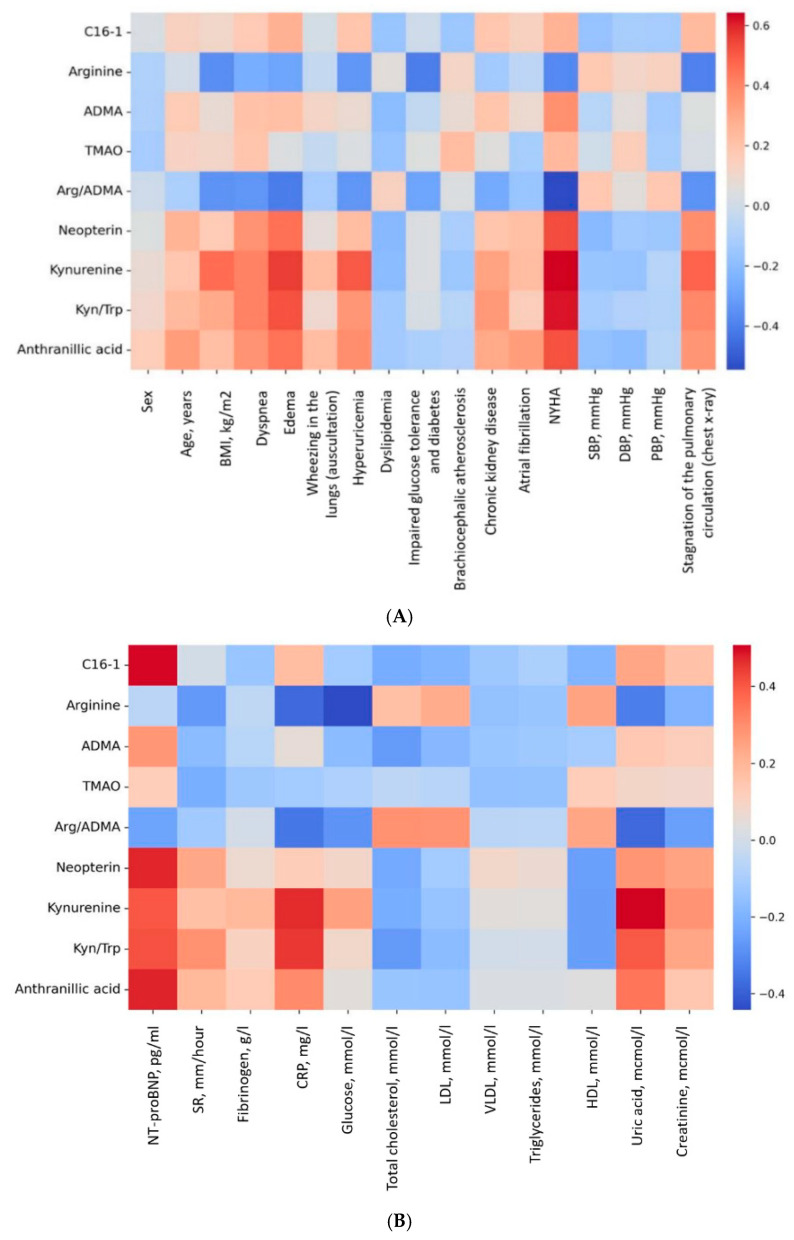
Correlations of the concentrations of metabolites (μM) in the HFpEF group with laboratory and instrumental data. (**A**) Correlations of the concentrations of metabolites with clinical and demographic data. (**B**) Correlations of the concentrations of metabolites with laboratory data. (**C**) Correlations of the concentrations of metabolites with EchoCG parameters.

**Figure 4 ijms-26-02102-f004:**
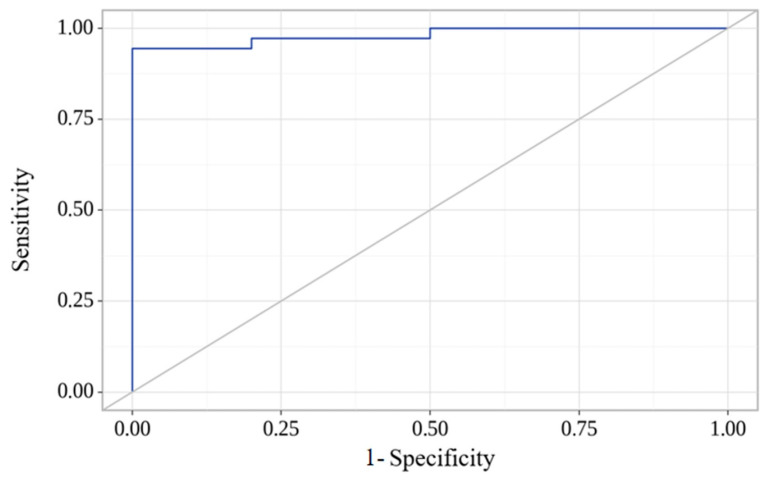
ROC analysis of the metabolomic panel.

**Table 1 ijms-26-02102-t001:** Demographic, hemodynamic parameters, risk factors, associated conditions, and target organ damage, treatment of study participants.

Parameters	Hypertension (*n* = 40)	HFpEF (*n* = 36)	*p*-Value
	Abs. (%) Me [Q_1_; Q_3_]	Abs. (%) Me [Q_1_; Q_3_]	
Sex, *n* (male)	17 (42.5)	13 (36.1)	0.569
Age, years	66.5 [60.5; 69.5]	69.5 [64.0; 73.0]	0.085
BMI, kg/m^2^	29.17 [25.86; 32.33]	34.24 [29.76; 38.36]	<0.001 **
HR, bpm	67.0 [60.0; 71.0]	76 [66; 90]	0.006 *
SBP, mmHg	140 [130; 160]	140 [120; 150]	0.379
DBP, mmHg	90 [85; 95]	80 [80; 90]	0.175
PBP, mmHg	50 [45; 70]	50 [48; 68]	0.468
Obesity, *n*	17 (42.5)	26 (72.2)	0.009 **
Dyslipidemia, *n*	40 (100)	34 (94.4)	0.221
Hyperuricemia, *n*	10 (25.0)	19 (52.8)	0.013 *
Impaired glucose tolerance and diabetes, *n*	15 (37.5)	17 (47.2)	0.391
Brachiocephalic atherosclerosis, *n*	17 (42.5)	13 (36.1)	0.569
Atrial fibrillation, *n*	13 (32.5)	20 (55.6)	0.043 *
Chronic kidney disease, *n*	13 (32.5)	19 (52.8)	0.074
Stroke, *n*	1 (2.5)	7 (19.4)	0.023 *
History of COVID-19, *n*	5 (12.5)	6 (16.7)	0.747
6-min walk test	-	II grade NYHA (50) III grade NYHA (50)	-
Congestion in the small circulation (chest radiography)	0	12 (33.3%)	<0.001 **
ACEI	30 (75.0)	18 (50.0)	0.024 *
ARB	10 (25.0)	18 (50.0)	0.024 *
BAB	22 (55.0)	29 (80.6)	0.018 *
Diuretics	18 (45.0)	25 (69.4)	0.032 *
MCRA	1 (2.5)	20 (55.6)	<0.001 **
Anticoagulants	12 (30.0)	21 (58.3)	0.013 *
Statins	26 (65.0)	20 (55.6)	0.400
Class III antiarrhythmics	6 (15.0)	4 (11.1)	0.740
Hypoglycemic drugs	7 (17.5)	12 (33.3)	0.111

Abbreviations: ACEI, angiotensin-converting-enzyme inhibitors; ARB, angiotensin II receptor blockers; BAB, beta-adreno-blockers; DBP, diastolic blood pressure; HR, heart rate; MCRA, mineralocorticoid receptor antagonists; PBP, pulse blood pressure; SBP, systolic blood pressure. * *p* ≤ 0.05; ** *p* ≤ 0.01.

**Table 2 ijms-26-02102-t002:** Laboratory and instrumental data of the study participants.

Laboratory Parameters	Hypertension (*n* = 40)	HFpEF (*n* = 36)	*p* Criteria
	Me [Q_1_; Q_3_]	Me [Q_1_; Q_3_]	
NT-proBNP, pg/mL	48.00 [35.25; 82.50]	946.31 [241.78; 2694.56]	<0.001 **
ESR, mm/hour	11 [8; 15]	15 [10; 20]	0.057
Fibrinogen, g/L	3.62 [3.14; 4.51]	3.81 [3.17; 4.55]	0.617
CRP, mg/L	1.7 [0.4; 3.0]	2.65 [1.81; 7.32]	0.018 *
Glucose, mmol/L	5.5 [5.2; 6.3]	5.8 [5.2; 7.0]	0.381
Total cholesterol, mmol/L	5.41 [4.70; 6.13]	4.76 [3.80; 5.41]	0.043 *
LDL, mmol/L	3.30 [2.62; 3.84]	2.96 [2.20; 3.67]	0.352
VLDL, mmol/L	0.64 [0.44; 0.84]	0.66 [0.50; 0.89]	0.561
Triglycerides, mmol/L	1.40 [0.96; 1.84]	1.48 [1.10; 1.93]	0.548
HDL, mmol/L	1.45 [1.22; 1.72]	1.06 [0.91; 1.49]	0.016 *
Uric acid, μmol/L	325.0 [282.4; 378.0]	387.5 [315.0; 453.9]	0.016 *
Creatinine, μmol/L	93.9 [83.3; 104.5]	92.3 [81.9; 105.4]	1.000
Echocardiographic parameters			
LV PW, cm	11.00 [10.00; 12.00]	12.00 [11.00; 13.00]	0.003 **
IVS, cm	11.00 [9.75; 12.00]	13.00 [11.75; 14.12]	<0.001 **
EDD, cm	4.8 [4.6; 5.1]	4.9 [4.6; 5.1]	0.751
LV RWT, cm	0.45 [0.41; 0.50]	0.50 [0.46; 0.55]	<0.001 **
LVMW, g	187.54 [160.88; 216.51]	233.75 [204.88; 272.74]	<0.001 **
LVMI, g/m^2^	106.56 [88.78; 128.81]	127.85 [111.50; 149.71]	0.002 **
LVEF, %	60 [57; 63]	57 [55; 62]	0.006 **
E/A	0.93 [0.69; 1.17]	0.70 [0.60; 0.80]	0.009 **
E/e’	8.6 [7.4; 10.1]	11.5 [10.4; 13.5]	0.004 **
LAV, mL	52.00 [45.00; 70.50]	69.00 [58.50; 88.00]	<0.001 **
LAVI, mL/m^2^	29 [25; 33]	34 [29; 41]	0.001 **
RAV, mL	46.00 [40.00; 57.00]	51.00 [42.00; 64.25]	0.214
PASP, mmHg	24 [21; 27]	29 [25; 35]	0.009 **

Abbreviations: CRP—C-reactive protein, EDD—end-diastolic dimension, E/A—E wave—A wave ratio, E/e’—E/e’ ratio, ESR—erythrocyte sedimentation rate, HDL—high-density lipoprotein, IVS—interventricular septum, LAV—left atrial volume, LAVI—left atrial volume index, LDL—low-density lipoprotein, LV PW—left ventricular posterior wall, LV RWT—left ventricular relative wall thickness, LVEF—left ventricular ejection fraction, LVMI—left ventricular mass index, LVMW—left ventricular myocardial mass, PASP—pulmonary artery systolic pressure, RAV—right atrial volume, VLDL—very low density lipoprotein. * *p* ≤ 0.05; ** *p* ≤ 0.01.

**Table 3 ijms-26-02102-t003:** The most important metabolites based on the conducted OPLS-DA analysis.

Metabolites	VIP Score	S-Plot	
		*p*	*p*-Corr
Melatonin	2.61864	−0.00339901	−0.578111
Indole-3-carboxaldehyde	2.45757	0.0592664	0.677344
Kynurenine	2.29294	−0.220714	−0.725247
Kynurenine/tryptophan ratio	2.23577	−0.00111338	−0.716972
Arginine/ADMA ratio	1.93259	0.0947818	0.660769
VMA	1.85746	−0.0960948	−0.621775
Anthranilic acid	1.82358	−0.0167127	−0.596247
Neopterin	1.81932	−0.0226305	−0.568105
Biopterin	1.81175	−0.00771421	−0.55114
C5:1	1.59271	0.000481496	0.405371
Indole-3-acrylic acid	1.49558	0.00440939	0.432195
Tyrosine	1.4454	0.0301244	0.448044
3-OH Kynurenine	1.44046	−0.170487	−0.437577
Histidine	1.43888	0.0201054	0.455949
Leucine	1.34359	0.0426002	0.429608
Arginine	1.33847	0.0437727	0.509689
3-Aminoisobutyric acid	1.33817	−0.283445	−0.418619
ADMA	1.31838	−0.00233046	−0.381152
Lysine	1.26998	0.0506475	0.441763
Methionine	1.24508	0.016927	0.417841
Tryptophan	1.17388	0.888168	0.371326
Threonine	1.1072	0.036057	0.373447
TMAO	1.03597	−0.0117037	−0.239662
Phenylalanine/tyrosine ratio	1.03222	−0.00240163	−0.314995

Abbreviations: ADMA—asymmetric dimethylarginine; C5:1—tiglylcarnitine; TMAO—trimethylamine-N-oxide; VMA—vanillylaminedelic acid.

**Table 4 ijms-26-02102-t004:** Concentrations of metabolites included in the diagnostic panel for HFpEF.

Metabolites	Hypertension (*n* = 40)	HFpEF (*n* = 36)	*p* Criteria
	Me [Q_1_; Q_3_]	Me [Q_1_; Q_3_]	
C16:1	0.016 [0.011; 0.018]	0.018 [0.015; 0.026]	0.010
Arginine	102.1 [83.6; 115.1]	80.1 [61.0; 93.9]	0.002
ADMA	0.398 [0.369; 0.471]	0.476 [0.434; 0.579]	<0.001
TMAO	2.60 [1.25; 4.95]	4.30 [2.15; 10.13]	0.026
Arginine/ADMA ratio	231.9 [196.0; 294.1]	154.9 [126.3; 203.0]	<0.001
Neopterin	5.18 [3.88; 7.44]	8.92 [6.52; 13.76]	<0.001
Kynurenine	585.7 [515.7; 741.9]	966.9 [727.0; 1237.1]	<0.001
Kynurenine/tryptophan	0.009 [0.007; 0.012]	0.017 [0.014; 0.023]	<0.001
Anthranilic acid	7.04 [5.35; 8.81]	9.38 [7.59; 11.83]	<0.001

C16:1—hexadecenoylcarnitine.

**Table 5 ijms-26-02102-t005:** ROC curve analysis of potential biomarkers of HFpEF.

Metabolites	AUC	95% CI	Sensitivity, %	Specificity, %	*p* Criteria
C16:1	0.671 ± 0.062	0.549–0.793	63.9	67.5	0.010
Arginine	0.709 ± 0.060	0.592–0.826	83.3	55.0	0.002
ADMA	0.722 ± 0.059	0.607–0.838	77.8	62.5	<0.001
TMAO	0.649 ± 0.063	0.524–0.773	44.4	80.0	0.026
Arginine/ADMA ratio	0.806 ± 0.051	0.706–0.906	83.3	70.0	<0.001
Neopterin	0.799 ± 0.052	0.698–0.901	58.3	85.0	<0.001
Kynurenine	0.876 ± 0.042	0.794–0.958	61.1	97.5	<0.001
Anthranilic acid	0.780 ± 0.054	0.674–0.885	75.0	67.5	<0.001
Kynurenine/tryptophan ratio	0.892 ± 0.039	0.815–0.968	83.3	82.5	<0.001

## Data Availability

The original contributions presented in this study are included in the article/Appendix A. Further inquiries can be directed to the corresponding author.

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
