# Peer review of "Metabolomic Panel for the Diagnosis of Heart Failure with Preserved Ejection Fraction"

_ijms, 2025, doi:10.3390/ijms26052102_

Round 1
Reviewer 1 Report
Comments and Suggestions for Authors
In their clinical study, Kozhevnikova et al. investigated the changes in pre-determined plasma metabolites in HFpEF patients compared to hypertensive patients. Their main findings are that i) 18 metabolites and 3 metabolite ratios changed significantly in HFpEF, and ii) 7 metabolites and 2 metabolite ratios were prognostic for HFpEF based on a model. The topic is interesting and has high clinical relevance. The study seems to have been carefully designed and executed.
1. Please list the 7 significantly changed metabolites and 2 ratios prognostic for HFpEF in the Abstract.
2. The Introduction is too long. Please shorten the paragraphs about galectin-3 and sST2 to half.
3. The authors mentioned that 72 metabolites were selected for screening. Please describe how these metabolites were selected in the Methods section.
4. Please change "Annotations" to "Abbreviations" in the Tables.
of HF with preserved ejection fraction
Reviewer 2 Report
Comments and Suggestions for Authors
The authors used targeted metabolomics to find metabolomic biomarkers for the diagnosis of heart failure with preserved ejection fraction (HFpEF). The study included 76 participants with hypertension, 36 of whom were diagnosed with HFpEF. There were 18 statistically significant differences in concentrations of metabolites and 3 differences in their ratios between HFpEF and hypertension groups. The prognostic model for detecting the possibility of HFpEF included seven metabolites and two ratios with ROC curve was 0,981 ± 0,017.
Although the study was limited with a small sample size ( Just 76 participants), The authors addressed this point at the Limitations section.
I recommend acceptance after a revision.
1- There were some significant differences in several factors, such as obesity (p=0.009), hyperuricemia (p=0.013), and history of acute cerebral circulatory failure (p=0.023) , which were more common in the HFpEF group. How do the authors adjust the results of the study so that the detected metabolites that significantly increased HFpEF group are associated to HFpEF and not to one of these factors that differ between the groups?
2- Page 10, I suggest moving the paragraph of the Annotations from the body of the manuscript to a section of abbreviations. And there is no need to write the annotations after each figure.
3- Page 11, May the authors explain how they calculated the z indices (−32,636, −31,380×𝐶1−0,232×𝐶2+24,914×𝐶3+0,475×𝐶4+0,049×𝑅1+0,349×С5+0,025×𝐶6+0,146×𝐶7+618,859×𝑅2 )
Reviewer 3 Report
Comments and Suggestions for Authors
The authors of the manuscript analyzed compounds that could become markers of heart failure with preserved systolic function HFpEF. The work is interesting and well planned. I have a few comments that I would like to consider:
- technical comments:
- figures - poor resolution. additionally figure 1 C - please change the scale range and use a non-linear scale. Please use color. Currently, the figure is illegible.
- in tables and text, please use the Greek letter "mi" - μ instead of the abbreviation mc... of course, this is formally correct, but why when it can be written correctly.
- comments on the text:
- the aim of such work is to present potential markers that facilitate diagnosis. A cardiologist specialist generally has no problem with diagnosis, but a family doctor who already takes care of you may have one. Hence, it is so important to propose a procedure. Please rank the markers from those most useful in the light of the study. It would be good to summarize in the conclusions
- the group of patients is relatively small, hence several of the following problems
+ diagnosis of obesity, for example - how was it made? based on BMI? body composition measurement? what method? MRI? The assessed factors should be addressed in the same way.
+ I lack the characteristics of the population in terms of the risk factors occurring
+ were the patients treated earlier? how? what drugs? could there be drug-marker interactions? Statistically, it would be difficult to assess the impact of individual markers because the population is small.
I hope that after appropriate modification, the work can be considered for publication.
Round 2
Reviewer 3 Report
Comments and Suggestions for Authors
The authors have been making corrections to the manuscript - I believe it can now be considered for publication.
Author Response
Dear Reviewer,
Thank you very much for taking the time to review this manuscript again.
Please find the response below and the corresponding revisions highlighted in the re-submitted file at page 8.
You are absolutely right and we agree that the analysis of the influence of risk factors that differ between groups is very important. However, multiple analysis of all identified significant metabolites is extremely voluminous, we would like to draw attention to the most interesting, in our opinion, results of the analysis for the factors included in the analysis.
" Multiple linear and variance analyses were performed to examine the effects of statistically significantly different risk factors and comorbidities between groups, such as BMI (p = 0.001) and obesity (p = 0.009), higher uric acid levels (p = 0.016) and hyperuricemia (p = 0.013), atrial fibrillation, and therapy. Body mass index had a significant effect on the levels of medium-chain acylcarnitines (p = 0.002), tyrosine (p = 0.019), histidine (p < 0.001), ADMA (p = 0.007), antranillic acid (p = 0.010), and arginine (p < 0.001). C5-1 (p = 0.002) and Arginine/ADMA ratio (p = 0.006) were associated with uric acid metabolism disorders. Drug therapy (diuretics) had little effect on the kynurenine/tryptophan ratio (p = 0.047)."
